# Specific Inhibition of VanZ-Mediated Resistance to Lipoglycopeptide Antibiotics

**DOI:** 10.3390/ijms23010097

**Published:** 2021-12-22

**Authors:** Vishma Pratap Sur, Aninda Mazumdar, Vladimir Vimberg, Tommaso Stefani, Ladislav Androvic, Lucie Kracikova, Richard Laga, Zdenek Kamenik, Katerina Komrskova

**Affiliations:** 1Laboratory of Reproductive Biology, Institute of Biotechnology of the Czech Academy of Sciences, BIOCEV, Prumyslova 595, 252 50 Vestec, Czech Republic; vishmapratap.sur@ibt.cas.cz; 2Laboratory for Biology of Secondary Metabolism, Institute of Microbiology of the Czech Academy of Sciences, Videnska 1083, 142 20 Prague, Czech Republic; aninda.mazumdar@biomed.cas.cz (A.M.); vladimir.vimberg@gmail.com (V.V.); tommaso.stefani@biomed.cas.cz (T.S.); kamenik@biomed.cas.cz (Z.K.); 3Department of Polymer and Colloid Immunotherapeutics, Institute of Macromolecular Chemistry of the Czech Academy of Sciences, Heyrovskeho namesti 2, 162 06 Prague, Czech Republic; androvic@imc.cas.cz (L.A.); woldrichova@imc.cas.cz (L.K.); laga@imc.cas.cz (R.L.); 4Department of Polymers, Faculty of Chemical Technology, University of Chemistry and Technology, Technicka 5, 166 28 Prague, Czech Republic; 5Department of Zoology, Faculty of Science, Charles University, Vinicna 7, 128 44 Prague, Czech Republic

**Keywords:** antibiotic resistance, benzimidazole, docking, *Enterococcus faecium*, ligands, lipoglycopeptide antibiotic, MD simulation, *Staphylococcus aureus*, teicoplanin VanZ

## Abstract

Teicoplanin is a natural lipoglycopeptide antibiotic with a similar activity spectrum as vancomycin; however, it has with the added benefit to the patient of low cytotoxicity. Both teicoplanin and vancomycin antibiotics are actively used in medical practice in the prophylaxis and treatment of severe life-threatening infections caused by gram-positive bacteria, including methicillin-resistant *Staphylococcus aureus*, *Enterococcus faecium* and *Clostridium difficile*. The expression of *vancomycin Z* (*vanZ*), encoded either in the *vancomycin A* (*vanA*) glycopeptide antibiotic resistance gene cluster or in the genomes of *E. faecium*, as well as *Streptococcus pneumoniae* and *C. difficile*, was shown to specifically compromise the antibiotic efficiency through the inhibition of teicoplanin binding to the bacterial surface. However, the exact mechanisms of this action and protein structure remain unknown. In this study, the three-dimensional structure of VanZ from *E. faecium EnGen0191* was predicted by using the I-TASSER web server. Based on the VanZ structure, a benzimidazole based ligand was predicted to bind to the VanZ by molecular docking. Importantly, this new ligand, named G3K, was further confirmed to specifically inhibit VanZ-mediated resistance to teicoplanin in vivo.

## 1. Introduction

Antibiotic resistance in bacteria represents a serious global threat requiring the search for new compounds to treat bacterial infectious diseases. At the same time, it is crucial to preserve the ability to use clinically accepted antibiotics, especially so-called “last resort” antibiotics, which are used for treating multiple drug-resistant infectious agents. The development of new therapeutics that modulate existing resistance mechanisms may increase the lifetime of already accepted antibiotics and overcome the spread of resistance [1,2,3,4,5,6,7].

*E. faecium* is a common infection in immune-compromised patients and is responsible for a significant percentage of healthcare-associated infections [8,9]. It is mostly harmless to healthy individuals, but it can cause a diverse range of infections in immunodeficient or elderly patients with severe consequences. The normal gut microbiota of patients are affected by the administration of broad-spectrum antibiotics, facilitating the outgrowth of antibiotic-resistant *E. faecium.* Such bacteria outgrowth may result in enterococcal infections via direct bacterial translocation from the gut to the bloodstream, or via fecal contamination leading to the urinary tract, surgical-site infections, or colonization of the skin [10]. The natural glycopeptide antibiotic, vancomycin (VAN) and the natural lipoglycopeptide antibiotic, teicoplanin (TEI) [11], have been used as ‘last resort’ antibiotics to treat *E. faecium* infections. These antibiotics bind to the D-alanyl-D-alanine (D-Ala-D-Ala) terminus of the bacterial peptidoglycan precursor [12], resulting in the inhibition of the cell wall synthesis and subsequent cell lysis. Due to TEI’s efficiency and much lower cytotoxicity in comparison to VAN, TEI has been preferentially used in patients with medical complications, including pediatric and immune-compromised patients.

The most common glycopeptide antibiotic resistance in *Enterococci* spp. is due to the reprogramming of the cell wall by enzymes encoded in the *vanA* glycopeptide antibiotic resistance gene cluster. The gene cluster encodes *vanHAX* genes that are responsible for production of peptidoglycan precursors, containing D-alanine-D-lactate instead of the D-Ala-D-Ala, reducing the affinity of glycopeptide antibiotics to the peptidoglycan [13]. The *vanA* gene cluster contains two additional genes: *vanY* and *vanZ*. VanY, a D,D-carboxypeptidase, eliminates D-Ala-D-Ala from peptidoglycan precursors, minimizing the number of primary binding sites for glycopeptide antibiotics. VanZ is independent of the peptidoglycan modification mechanism, and decreases the sensitivity of bacteria to lipoglycopeptide antibiotics, including TEI, but not VAN [2,14,15,16]. The *vanZ* orthologs are also present in the genomes of pathogenic bacteria like *Streptococcus* spp. [15], *Clostridium* spp. [16], *Bacillus* spp. [17] and *Enterococcus* spp. [15], and have been shown to contribute to lipoglycopeptide antibiotic resistance. However, the mechanism of VanZ-mediated resistance has not been thoroughly studied.

In this paper, the structure of VanZ was solved in silico. Based on this structure, we have predicted in silico that a derivative of benzimidazole, N-(3-(1H-tetrazol-5-yl) phenyl)-1H-benzimidazole-4-carboxamide (G3K), can be docked to VanZ. A molecular dynamics simulation was performed with the best pose of VanZ to confirm the stability of VanZ–G3K and validate the molecular docking results. Finally, G3K activity was validated by an in vivo experiment, demonstrating the specific inhibition of VanZ-mediated resistance to TEI.

## 2. Results

### 2.1. VanZ Modeling and Prediction of the VanZ Interaction with a Derivative of Benzimidazole

In this study, we used the I-TASSER web server to predict the 3D structure of VanZ (Figure 1). Ten templates taken into consideration for the modeling had a Z-score > 1 (Appendix A). Five predicted models (Figure 1) and Appendix A, had a C-score between −2.13 and −5.0 (Appendix A). The predicted structure of VanZ had a high confidence score for the majority of the amino acid sequence (Appendix A). The predicted normalized B-factor value was mostly in the negative region of the axis (Appendix A). The overall results suggested that the model with the highest C-score (−2.13) was the best example for follow up studies. A Ramachandran plot obtained from Procheck (Appendix A), showed that 90.3% of the residues were in the most favored regions and 6.9% and 1.4% were in the additional and generously allowed regions, respectively. Meanwhile, only 1.4% were in the disallowed region.

Molecular docking was done first to determine the binding pocket using CASTp. It was used to determine the binding pocket solvent accessible (SA) surface area and volume. The prediction showed the location of the binding pocket in the interior region of the protein as well as on the surface (Figure 2A). The binding pocket volume (SA) and area (SA) of VanZ were 318.192 Å^3^ and 706.856 Å^2^, respectively. This signifies the space for the optimum ligand binding. The best-ranked (based on pocket size and highest number of participating amino acids) pocket for docking analyzed by the CASTp server was selected for the VanZ–G3K molecular docking (Figure 2A). With the PyRx AutoDock vina module, the benzimidazole derivative G3K ligand was predicted to dock with the VanZ protein using the specific target pocket, as predicted by CASTp (Figure 2A). Out of the nine predicted poses, the best docking score for VanZ–G3K was −7.7 kcal/mol (Figure 2B). The rest of the poses had scores between −6.9 and −7.6 kcal/mol (Appendix A). The model with the lowest binding score was used for further studies.

A BIOVIA Discovery Studio protein–ligand interaction module-based analysis suggested the probable amino acids participating in the interaction with G3K. The amino acids that interacted with G3K are Arg52, Thr48, Leu144, Val147 and Val151 (Figure 2C). Whereas Arg52 interacted with G3K via the hydrogen bond and Pi-Alkyl bond, Leu144 and Val147 interacted via the Pi-Alkyl bond, Thr48 interacted via the Pi-Sigma bond and hydrogen bond, and Val151 interacted via the Pi-Sigma bond.

The Protter web-based tool helped to identify the five-membrane domain present in the VanZ protein, (Appendix A), which agreed with the membrane model obtained from CHARMM-GUI (Figure 3A), where the membrane embedded with VanZ is represented from a close side view (Figure 3B), and top view (Figure 3C).

### 2.2. Molecular Dynamic Simulation of VanZ–G3K and Analysis

An all-atom molecular dynamic simulation was conducted with the best docking pose of the VanZ–G3K. The dynamic features of this interaction were analyzed by the following parameters: gmx RMSD, gmx RMSF, gmx Rg plot, gmx H-bonds, gmx SASA and gmx interaction energy.

To determine the complex conformation stability of VanZ docked with G3K (−7.7 kcal/mol), the backbone root mean square deviation (Cα-RMSD) was computed (Figure 4A). The result showed that the RMSD trajectory of VanZ–G3K was equilibrated during 0–20 ns and remained steady until 70 ns with a RMSD value of ~6 ± 0.1 Å. Afterward, it showed a small drift from 70 ns to 85 ns, and finally the trajectory became stable again from 85 ns to the end of the simulation at ~7 ± 0.2 Å (Figure 4A). This indicates a very stable structural complexity of the VanZ–G3K complex. It is also stable enough to allow the free conformational inter-conversion of the ligand.

The RMSF plot of VanZ–G3K, shows that the amino acid residues belonging to the N-terminal and C-terminal had an average atomic fluctuation of ~4.0 Å, whereas the loops had an average atomic fluctuation of ~5.0 Å (Figure 4B). The residue between 20 and 40 amino acids showed a higher fluctuation. In the 100 ns simulation, the C-terminal showed a slightly higher flexibility than the N-terminal. However, the conformational dynamics of the stable secondary structure, α-helices and β-sheets (interacting protein residues with the ligand compounds) remained stable during the entire simulation process.

To examine the structural compactness and integrity of VanZ–G3K-bound complexes, the radius of gyration (Rg) was calculated [18,19]. It was observed (Figure 4C) that the structure of VanZ–G3K was stabilized around an Rg value of 18 ± 0.1 Å. Two structural drifts were also visualized around 10 ns reporting an Rg value dropping from 18 ± 0.1 Å to 17.9 ± 0.1 Å, and this was maintained for 70 ns. Meanwhile, the second structural drift was around 90 ns, and the Rg value shifted from 17.9 ± 0.1 Å to 17.0 ± 0.1 Å (Figure 4C). The structural compactness of the VanZ–G3K ligand, as calculated by the Rg analysis, was stabilized at 17.9 ± 0.1 Å. The structural drift and change in the Rg value was much lower, which clearly indicates the high compactness of the VanZ protein in the presence of G3K (Figure 4C).

Furthermore, the H-bond between VanZ and the G3K complex was computed by a time evolution plot of the hydrogen bond occupancy (H-bonds). In the case of VanZ–G3K interactions, initially six H-bonds were detected; however, over time the number of H-bonds reduced. Finally, at 100 ns two H-bonds were detected, which still supports our docking interaction analysis data (Figure 4D).

The computation of the solvent-accessible surface area (SASA) helped us to study the changes in the structural features of the VanZ–G3K complex. VanZ occupied with G3K showed an average SASA value of 96 ± 2 nm^2^ (Figure 4E). In the VanZ–G3K interaction no prominent fluctuations in the protein-accessible area were detected, which is an indication of an insignificant change in orientation on the protein surface. 

For VanZ–G3K, the average value of the short-range electrostatic (Coul-SR) interaction energy was −22.78 ± 1.8 kJ/mol, and the van der Waals/hydrophobic (LJ-SR) interaction energy was −86.9087 ± 5.3 kJ/mol (Figure 4F). The Coul-SR and LJ-SR interaction energies between the VanZ–G3K complex explained promising electrostatic as well as hydrophobic interactions. The hydrophobic interaction played an important role in comparison to the electrostatic interactions, as suggested by the short-range interaction energy calculation [20,21], in stabilizing the complex.

Altogether our docking and modular dynamic (MD) simulation studies suggested that G3K had a high binding affinity and stable interaction with VanZ. We decided to synthesize the G3K and to investigate the effect of the G3K on VanZ function in vivo. 

### 2.3. Synthesis and Characterization of G3K

The G3K synthesis was performed in two synthetic steps (details are presented in Section 4.2) (Figure 5). The first steps involved a (3+2)-dipolar cycloaddition reaction between amino benzonitrile and sodium azide in the presence of TEA∙HCl, yielding the tetrazole-group-containing intermediate 3-(1H-tetrazol-5-yl)aniline. The next step involved the condensation of 3-(1H-tetrazol-5-yl) aniline with 1H-benzimidazole-4-carboxylic acid in the presence of the uronium-based coupling reagent COMU and TEA as a base. The pure product N-(3-(1H-tetrazol-5-yl) phenyl)-1H-benzimidazole-4-carboxamide (G3K) was obtained by purifying the reaction mixture by flash chromatography on a C-18 reverse-phase silica gel column. The structure and purity of the G3K was verified by ^1^H-NMR and HPLC analyses, as documented in Appendix A. Finally, the G3K chromatogram obtained from the HPLC analysis showed a purity ≥99%. For these reasons, the obtained compound was used for the following in vivo investigation.

### 2.4. In Vivo Validation of VanZ–G3K Interaction

The synthesized G3K was first evaluated on four gram-positive strains (*C. difficile*, *S. pneumoniae*, *E. faecium*, *S. aureus*) (Appendix A) to study its antibacterial potential. Different concentrations (0.5, 1, 2, 4, 8, 16, 32, 64, 128, 256 and 512 µg/mL) of G3K were used to study its effect on bacterial growth after overnight incubation. The exposure of these strains to the G3K did not affect bacterial growth (Appendix A). Next, the G3K activity on VanZ was checked in vivo, using *S. aureus* RN4220 strain, with the heterologous expression of VanZ [22]. *S. aureus* does not contain *vanZ* in the genome. Thus, G3K would affect only heterogeneously expressed *vanZ*. 

*S. aureus* RN4220 strains heterogeneously expressing VanZ_TEI (*vanZ* encoded in the *vanA* resistance gene cluster) or VanZ_GEN (*vanZ* encoded in *E. faecium* genome) or bearing an empty plasmid (details in Section 4.5.2.), were exposed to the different concentrations of the TEI in the presence of varying concentrations (as mentioned above) of the G3K. Addition of the G3K increased the sensitivity of the bacteria, expressing VanZ, to the TEI. The inhibitory effect was more profound in the case of *vanZ* taken from the *vanA* resistance gene cluster than in the case of *vanZ* taken from the genome (Figure 6).

In the next experiment, the question addressed was whether G3K could increase the sensitivity of *E. faecium* clinical isolate (bearing the *vanA* resistance gene cluster) to TEI. Strain E1131, resistant to TEI (MIC > 1024µg/mL), decreased the sensitivity to TEI after the addition of the G3K compound to the growth medium, as shown in Figure 7. 

Finally, the effect of the G3K on the genomically encoded *vanZ*-mediated sensitivity to TEI in *S. pneumonia* was investigated. For controls we used *S. pneumoniae* with knockout and complemented with *vanZ*. The positive control strain (Δ*vanZ*) showed a high sensitivity towards TEI (MIC = 0.00156 µg/mL). The incubation of wild types (WT) and *vanZ*-reverted strains with G3K increased the sensitivity of the strains to TEI to the level seen in Δ*vanZ::vanZ* (Figure 8), demonstrating the direct effect of G3K on VanZ-mediated sensitivity to TEI. 

Altogether, the experiments performed showed that both results from the in silico and in vivo are in good agreement with each other. The in vivo experiment confirmed that G3K interacts with VanZ. This interaction—based on several types of bonds, both covalent and hydrogen—specifically inhibited VanZ-mediated resistance towards TEI, as was predicted in silico. G3K itself had no independent antibacterial activity on bacterial growth under laboratory growth conditions, whether the VanZ was encoded in the genome of the bacteria or not. This is in good agreement with previous results that demonstrated no effect of *vanZ* knockout in *S. pneumoniae* of *vanZ* heterologous overexpression in *S. aureus* on bacterial growth [22]. However, G3K managed to increase the efficiency of TEI against clinical isolate *E. faecium*, encoding *vanZ* in the *vanA* resistance gene cluster, and significantly decrease resistance to TEI in *S. pneumoniae*, encoding *vanZ* in its genome.

## 3. Discussion

Understanding the 3D structure of a protein is important for the prediction of its function. The analysis of a protein structure is a time-consuming and expensive process, and it can be difficult to obtain an accurate structure using protein crystallography or NMR. In silico software platforms such as MODELLER, AlphaFold, I-TASSER and SWISS-MODEL represent excellent tools to theoretically predict 3D protein structures that can then be experimentally verified. We used the in silico tool I-TASSER, to perform a de novo protein structure prediction of VanZ. VanZ is encoded in the *vanA* resistance gene cluster from *E. faecium*, which can specifically develop resistance to lipoglycopeptide antibiotics. In silico tools predicted that a benzimidazole derivative can interact with VanZ.

Benzimidazole-derived drugs such as omeperazole, and rabeprazole act as inhibitors of proton pumps and are used to treat stomach ulcers, while thiabendazole and albendazole inhibit tubulin polymerization and are used as anthelmintic drugs [23]. Hameed P, Shahul, et al. also showed in their study that benzimidazoles inhibit novel mycobacterial DNA gyrases [23]. One of our previous studies focused on the ruthenium Schiff base benzimidazole coordination complex, which showed significant antibacterial activity against different variants of *Staphylococcus aureus* [4,24]. A benzimidazole-based ligand has already demonstrated proficiency as an inhibitor of CTX-M-9 class A extended spectrum β-lactamase (ESBL) expressed in *Escherichia coli*, showing a binding affinity (Ki) of 1.3 μM [25]. Moreover, CTX-M-9 tetrazole moiety, combined with other pharmacophores, has the ability to enhance the antibacterial activity of the resulting tetrazole hybrid molecule against both sensitive and resistant bacterial strains. In fact, tetrazole–imidazole hybrids are reported to have potent to moderate activities (24–124 μg/mL) against many bacterial strains such as, *S. aureus*, *B. subtilis*, *E. coli*, *S. thyphimurium* and *C. albicans*, sometimes comparable to the positive control. In this study, a molecular docking study of VanZ and G3K interaction suggested an extremely high binding affinity of G3K with VanZ (−7.7 kcal/mol). Our study revealed that the G3K and VanZ interaction possesses both covalent bonding and hydrogen bonding, which strongly suggests an extremely high binding affinity between VanZ and G3K. Furthermore, MD simulation results explained several parameters of the VanZ–G3K structural complex and its stability. The RMSD value of ~7 ± 0.2 Å after 100 ns simulation without fluctuations (Figure 4A) indicated a very stable VanZ–G3K structural complexity and structural stability in the presence of G3K. It was stable enough to allow free conformational inter-conversion of the G3K. The RMSF plot of VanZ–G3K showed the C-terminal was slightly more flexible than the N-terminal. However, the conformational dynamics of a stable secondary structure, α-helices and β-sheets (protein residues interacting with the ligand compounds) remained stable during the entire simulation process. The average atomic fluctuations were measured using RMSF plots and clearly indicated that the benzimidazole-based G3K was well accommodated at the binding pocket of VanZ with advantageous stable molecular interactions with stable protein residues. The conformational stability of VanZ–G3K was evaluated by the radius of gyration (Rg), a parameter used by computational biologists to describe the structural compactness of proteins. The structural drift and change in the Rg value clearly indicate the high compactness of the VanZ protein in the presence of G3K. H-bonds have a crucial role in ligand binding. The results showed two H-bonds that agreed with the docking interaction results (Figure 4D). The SASA investigation of VanZ–G3K suggested no significant changes in the conformational dynamics of VanZ in the presence of the G3K ligand. Our MD simulation studies suggested a highly stable structural complexity of VanZ, high structural compactness, high level of flexibility and hydrophobicity in the presence of G3K. Importantly, the amount of LJ-SR interaction energy strongly suggested a high binding affinity between VanZ and G3K. Our in silico molecular docking and MD simulation studies were supported by in vivo experiments that demonstrated that G3K inhibited VanZ-mediated bacterial resistance to TEI. 

The in vivo results suggested that G3K was an effective inhibitor of not only VanZ encoded in the *vanA* glycopeptide antibiotics resistance gene cluster, but also of VanZ encoded in the *S. pneumoniae* genome. These VanZ proteins are not conserved in the amino acid sequence; however, these proteins have a similar membrane topology with five transmembrane domains [22]. This suggests that independently of the VanZ origin, the mode of action of the proteins can be the same, at least in relation to the lypoglycopeptide antibiotic resistance, and both types of VanZ can be inhibited by G3K. 

In several cases, ORFs coding for VanZ have been annotated as phosphotransbutyrylases. Although automatic annotation is not sufficient to predict the real enzymatic activity of VanZ, cell membrane modification by VanZ could provide an explanation for the decreased lipoglycopeptide antibiotic binding to the VanZ-carrying bacterial cells and explain the association of VanZ proteins with diverse cell processes [21]⁠. 

The acquisition and dissemination of resistant *vanZ* genes in *S. aureus* may become a critical problem. The encoding *vanZ* gene, as part of the *vanA* gene cluster, has occasionally been transferred from *enterococci* to *S. aureus* via transposons, resulting in highly vancomycin-resistant strains [26]⁠. Although the frequency of such an event appears to be low, due to the high fitness cost of *vanHAX*-mediated resistance in *S. aureus*, vancomycin-resistant staphylococcal strains may represent precursors for the generation of vanZ-bearing mobile genetic elements that can interfere with the action of semisynthetic lipoglycopeptide antibiotics. 

## 4. Materials and Methods

### 4.1. VanZ Protein Structure Prediction and Validation

The VanZ protein sequence (*Enterococcus faecium EnGen0191*) was obtained from UniProt (https://www.uniprot.org/uniprot/Q06242, accessed on 19 May 2021). Thereafter, the I-TASSER webserver (http://zhanglab.ccmb.med.umich.edu/I-TASSER/registration.html, accessed on 20 May 2021was used to obtain the 3D structure of VanZ. There are normally four steps involved in the I-TASSER server for structural modeling and prediction (https://zhanggroup.org/I-TASSER/about.html, accessed on 20 May 2021). 

The first step is the template identification using LOMETS, a meta-server threading approach which predicts the templates based on Z-scores (ideally > 1). The second step is to run Monte Carlo replica-exchange simulations to obtain the low-energy states and cluster centroids. The third step is a re-fragment assembly simulation to remove the steric clashes and atomic-level structure refinement of the cluster centroids global topology. The final stage is the structure-based function annotation by combining global and local search results (template model (TM) score [0, 1]; a higher score shows greater reliability). Following these steps, I-TASSER predicts the top five 3D models for each query protein sequence (C score [−5, 2]; a higher score proves there is a high level of confidence in the model) [27]. Furthermore, the protein structure was validated by a Ramachandran plot using the PROCHECK web server and by the Protter webserver to determine the transmembrane domain [28,29,30]. Finally, CHARMM-GUI was used to develop the membrane with the VanZ for visualization and better understanding of the protein [31]. 

### 4.2. Chemical Synthesis of G3K and Characterization

#### 4.2.1. Chemicals

3-Aminobenzonitrile, 1-Cyano-2-ethoxy-2-oxoethylidenaminooxy) dimethylamino-morpholino-carbenium hexafluorophosphate (COMU), sodium azide, triethylamine (TEA) and triethylammonium chloride (TEA∙HCl) were purchased from Merck, Czech Republic. 1H-Benzimidazole-4-carboxylic acid was obtained from Combi-Blocks, CA, USA. All solvents were of HPLC grade and dried over a layer of activated molecular sieves (4 Å) before use.

#### 4.2.2. Synthesis

Synthesis of N-(3-(1H-tetrazol-5-yl)phenyl)-1H-benzimidazole-4-carboxamide (G3K) was prepared in two synthetic steps (Figure 5). First, a mixture of 3-aminobenzonitrile (583 mg, 1.0 eq.), sodium azide (417 mg, 1.3 eq.) and TEA∙HCl (950 mg, 1.4 eq.) were dissolved in toluene (10 mL) and stirred at 100 °C for 28 h. The reaction mixture was then extracted into H_2_O (2 × 15 mL), and the aqueous fractions containing the product were combined and acidified with 35% HCl to pH 2. The resulting white precipitate was filtered off and dried in vacuo to give 684 mg (86%) of 3-(1H-tetrazol-5-yl)aniline. ^1^H-NMR (DMSO-d6, 400.13 MHz): δ 7.53 (m, 1H, Ar), 7.70 (m, 1H, Ar), 8.05 (m, 2H, Ar), 11.22 (br s, 2H, NH2). ^13^C-NMR (DMSO-d6, 100.61 MHz): δ 120.8, 125.0, 125.3, 125.8, 130.8, 134.7 and 155.3.

In the next step, TEA (311 μL, 1.0 eq.) and COMU (1.051 g, 1.1 eq.) were added to a solution of 1H-benzimidazole-4-carboxylic acid (362 mg, 1 eq.) in dimethylacetamide (DMAc, 10 mL), and the mixture was stirred at room temperature for 5 min. The next portion of triethylamine (466 μL, 1.5 eq.) and 3-(1H-tetrazol-5-yl) aniline (359 mg, 1 eq.) were then added and allowed to react at room temperature for 1 h. The precipitate formed was filtered off, and the product containing liquid residue was purified by flash chromatography on a C-18 reverse-phase silica gel column using a linear gradient of water-acetonitrile (0–100%). During the evaporation of acetonitrile, the solid product precipitated out of solution. The precipitate was filtered off and dried in vacuo to give 111 mg (16%) of resulting N-(3-(1H-tetrazol-5-yl)phenyl)-1H-benzimidazole-4-carboxamide. ^1^H-NMR (DMSO-d6, 400.13 MHz): δ 7.45 (m, 1H, Ar), 7.65 (m, 1H, Ar), 7.78 (m, 1H, Ar), 7.87 (m, 1H, Ar), 8.05 (m, 2H, Ar), 8.57 (s, 1H, Ar), 8.64 (s, 1H, Ar) and 12.14 (br s, 1H NH).

#### 4.2.3. Nuclear Magnetic Resonance (NMR) Spectroscopy

The synthesized chemical structures were investigated by ^1^H and ^13^C NMR spectroscopy in deuterated solvents on a Bruker DPX spectrometer (Bruker, Billerica, MA, USA) operating at 400.13 MHz. Both ^1^H and ^13^C spectra were calibrated to the signal; the internal standard was TMS δ = 0.00 MHz.

#### 4.2.4. High-Performance Liquid Chromatography (HPLC) 

The purity of chemicals synthesized in this study was verified on an HPLC system (Shimadzu, Japan) equipped with internal UV-VIS diode array detector (SPD-M20A) using a reversed-phase column Chromolith High Resolution RP-18e (Merck, Kenilworth, NJ, USA) with a linear gradient (0–100%) of water-acetonitrile containing 0.1% TFA at a flow rate of 2.5 mL/min.

### 4.3. Protein Pocket Analysis and Molecular Docking

The active site of the VanZ was predicted using CASTp (http://sts.bioe.uic.edu/castp/calculation.html, accessed on 22 May 2021). Based on possible solvent accessible surface volume and area the ligand-binding pockets were ranked [32].

In this study, we performed protein–ligand docking to obtain the binding pose and binding energy using AutoDock Vina 1.1.2 in PyRx 0.8 software. The ligand and protein files were uploaded to PyRx as ligand and macromolecules. These files were converted to PDBQT files for dockings. The binding pocket with the best possible size and binding residues was selected. The active site of the ligand was selected, and it was enclosed within a 3D affinity grid box. The grid box, which was centered to cover the active site residues, had the following dimensions: x = 68.41 Å, y = 61.57 Å and z = 64.51 Å. The exhaustiveness of the protein was set at eight. This provided nine predicted poses for the G3K with VanZ. The binding energies were recorded using Microsoft Excel (Office Version). The search space included the entire 3D structure of the VanZ. The initial visualization and analysis of the VanZ–G3K docking was performed by Chimera 1.15. Additionally, detailed analysis of the interacting amino acid and ligand was performed on BIOVIA Discovery Studio Visualizer [33].

### 4.4. Molecular Dynamic Simulation (MDS) Analysis

The best VanZ–G3K complex was chosen for MDS with the lowest binding energy and best docked pose. The binding interaction was also used for the MDS studies. The simulation was performed using the GROMACS 2020 package using an Ubuntu Linux system. The charmm36all atom force field was utilized for the molecular system by using semi-empirical, empirical and quantum-mechanical energy functions. The CGenff server (http://kenno.org/pro/cgenff/, accessed on 25 May 2021) was used to generate the input ligand files (topology and parameter files). To neutralize the system, a TIP3P water model was used to introduce the counter ions to the solvent. The protein was equilibrated, and the steepest descent minimization algorithm was used to perform energy minimization at 50,000 steps, accompanied by a conjugant gradient. For x, y and z directions, the PBC condition was defined, and simulations were performed at 300 K. To constrain hydrogen, PME (particle mesh Ewald)-treated long-range electrostatic forces and all bonding involved, the SHAKE algorithm was applied. Thereafter, the system was heated gradually at 300 K with 2 fs time step using 100 ps in the canonical ensemble (NVT) MDS. In the case of isothermal–isobaric ensemble (NPT) MDS, there was a 2 fs time step for one atom using 100 ps, and the atoms were relaxed at 300 K. The MDS for the system was conducted at 100 ns with a time step of 2 fs for 50,000,000 steps (100 ns) after equilibrating the system at the desired temperature and pressure. Energies and coordinates were saved every 10 ps for analysis.

The trajectory analysis module integrated in the GROMACS 2020.01 simulation package, VMD, qtgrace, python3 and Chimera software were used to analyze all trajectories. The trajectory files were analyzed by: GROMCAS tools; gmxrmsf; gmx gyrate; gmxhbond; gmxrmsd; gmxcovar; gmxsasa; gmx energy for extracting the graph of root-mean square fluctuations (RMSFs); radius of gyration (Rg); hydrogen bond; root-mean square deviation (RMSD); and solvent-accessible surface area (SASA) [33].

### 4.5. In Vivo Investigation

#### 4.5.1. Bacterial Strains

The gram-positive bacterial strains used to study G3K’s antibacterial effects were *C. difficile*, *E. faecium* (clinical isolates were provided by the General Hospital of Charles University) and *S. pneumonia* R6 code for *vanZ* inside the genome. VanZ from the *vanA*-phenotype gene cluster and *vanZ* encoded in *E. faecium* genome were heterologously expressed in *S. aureus* N4220, which was used to evaluate the activity of G3K. 

#### 4.5.2. Activity of G3K In Vivo against VanZ Expressing Bacterial Strains

##### G3K’s Effect against Gram-Positive Bacteria

Increasing amounts of G3K (0.5, 1, 2, 4, 8, 16, 32, 64, 128, 256 and 512 µg/mL) were added to *C. difficile*, *E. faecium*, *S. pneumonia* R6 and *S. aureus* RN4220 (the last of which does not have *v**anZ*) at the start of the bacterial growth in 96 well plates. After overnight incubation at 37 °C, the absorbance of the bacteria was measured at 600 nm (Tecan Infinite MNano+).

##### The Effect of G3K in Combination with TEI on VanZ-Mediated Teicoplanin Resistance in *S. aureus*

*S. aureus*, heterogeneously expresses VanZ_TEI or VanZ_GEN or bears an empty plasmid. VanZ from the *vanA*-phenotype gene cluster (VanZ_TEI), *vanZ* encoded in *E. faecium* genome (VanZ_GEN) and the plasmid with VanZ were heterogeneously expressed in *S. aureus*. They were grown in the presence of increasing amounts of G3K (0, 2, 16, 128 and 512 µg/mL) and/or TEI (0, 0.125, 0.25, 0.5, 1, 2, 4, 8 and 16µg/mL). After overnight incubation at 37 °C in 96 well plates, the absorbance was measured at 600 nm (Tecan Infinite MNano+). We have normalized the absorbance values of the bacterial grown in the presence of G3K and/or TEI with the absorbance of the bacterial grown in the absence of the compounds.

##### G3K in Combination with TEI against *S. pneumonia* Strains and Clinical Variants of *E. faecium* and Susceptibility Testing

Three clinical strains of *E. faecium* E1131 were selected with clinically confirmed *vanA* resistance phenotypes. For *S. pneumoniae* R6 there were three variant wild types (WT), Δ*vanZ* (*vanZ* knockout) and *vanZ*-reverted strains obtained as described in our previous publication [22]. These *E. faecium* and *S. pneumonia* R6 strains were grown in the presence of increasing amounts of G3K (0–512µg/mL) or TEI (0–1024µg/mL). After overnight growth in 96 well plates at 37 °C, the absorbance was measured at 600 nm (Tecan Infinite MNano+). The minimal inhibitory concentration (MIC) was measured according to the EUCAST standard methodology by the broth micro-dilution method, and interpretation was performed as stated using the EUCAST clinical breakpoint (https://eucast.org/clinical_breakpoints/ accessed on 10 July 2021) [22]. 

## 5. Conclusions

The overall study was performed to investigate a ligand with inhibitory effects towards the VanZ protein from the *vanA*-phenotype gene cluster. The in silico-based approach showed that the benzimidazole derivative G3K had a good binding affinity toward the VanZ. Following these results, we synthesized the G3K compound and demonstrated in vivo that G3K inhibits VanZ-mediated resistance to TEI. 

## Figures and Tables

**Figure 1 ijms-23-00097-f001:**
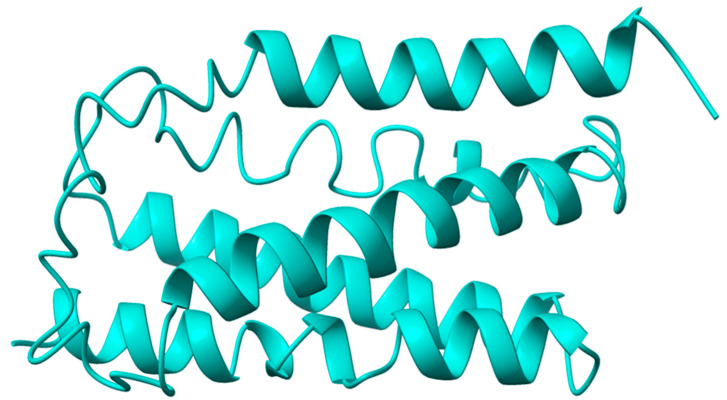
I-TASSER-server-based predicted structure of the VanZ protein. The prediction was repeated in the same server three times.

**Figure 2 ijms-23-00097-f002:**
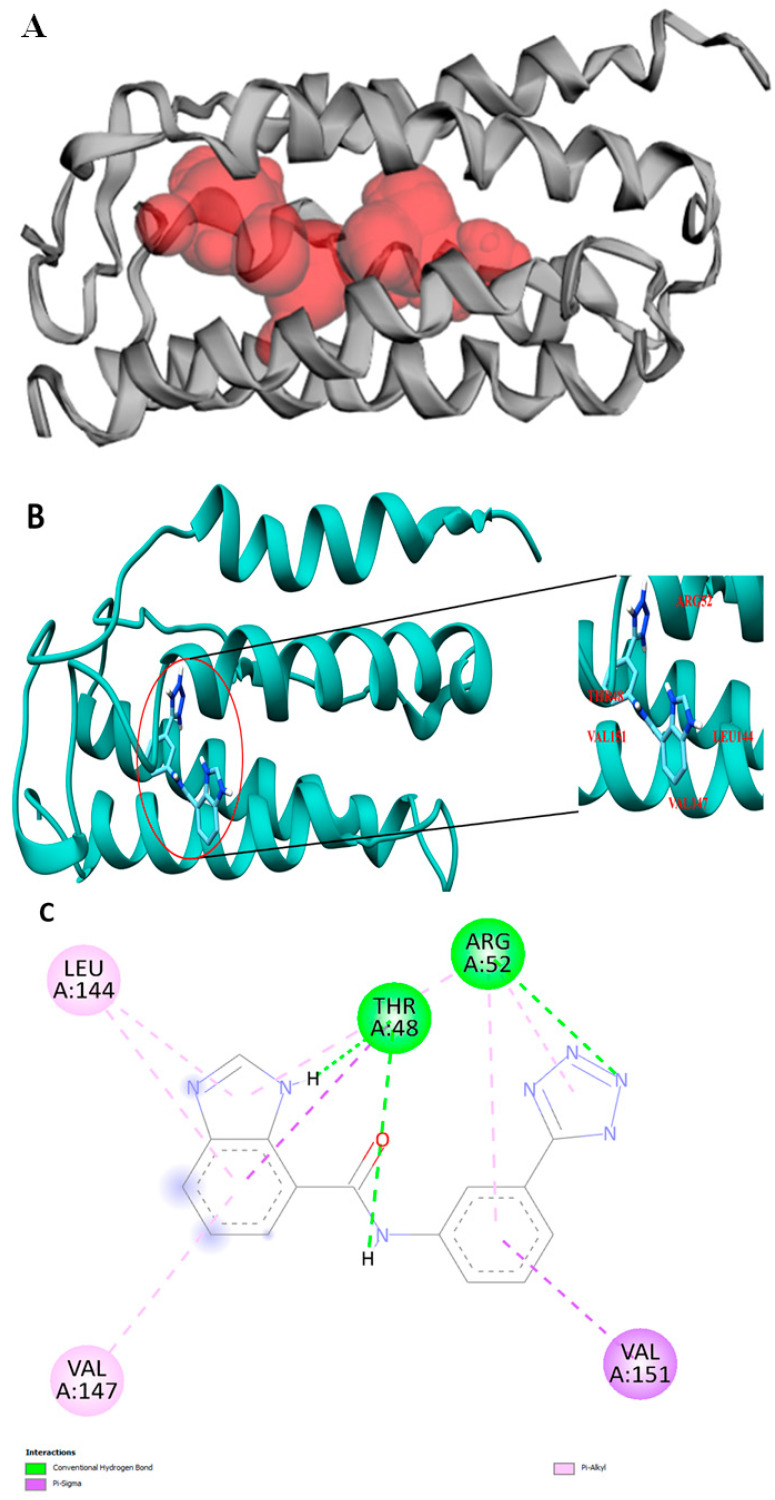
Docking pocket and representation of VanZ and G3K interaction. (**A**): Binding pocket predicted by CASTp. (**B**): TheVanZ–G3K docking pose. The amino acids interacting with G3K are marked. (**C**): A 2D representation showing the interacting amino acids, Arg52, Thr48, Leu144, Val147, and Val151 of VanZ with G3K ligand. Pocket prediction, docking and interaction analysis experiments were repeated three times each.

**Figure 3 ijms-23-00097-f003:**
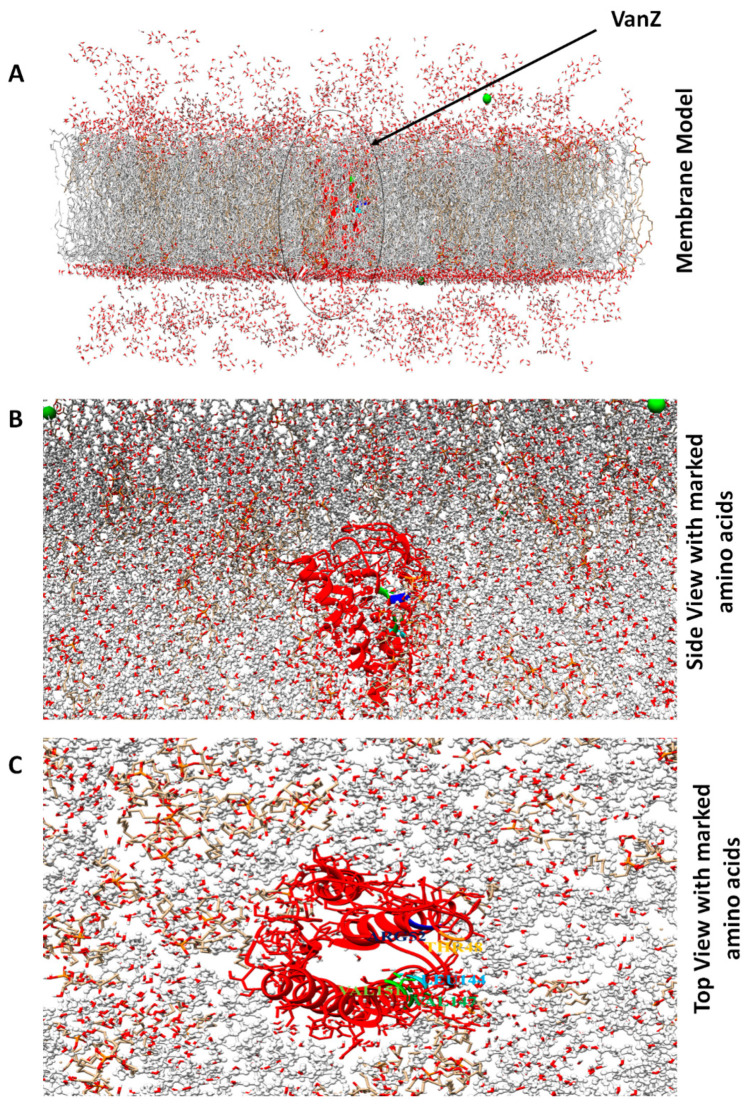
Bacterial protein-membrane model prepared by the CHARMM-GUI web-based modeling platform. (**A**): The VanZ protein is embedded in the bacterial membrane system. (**B**): VanZ (close side view) with color-marked interacting amino acids embedded in the bacterial membrane system. (**C**): VanZ (top view) with color-marked interacting amino acids initiating in G3K ligand interaction.

**Figure 4 ijms-23-00097-f004:**
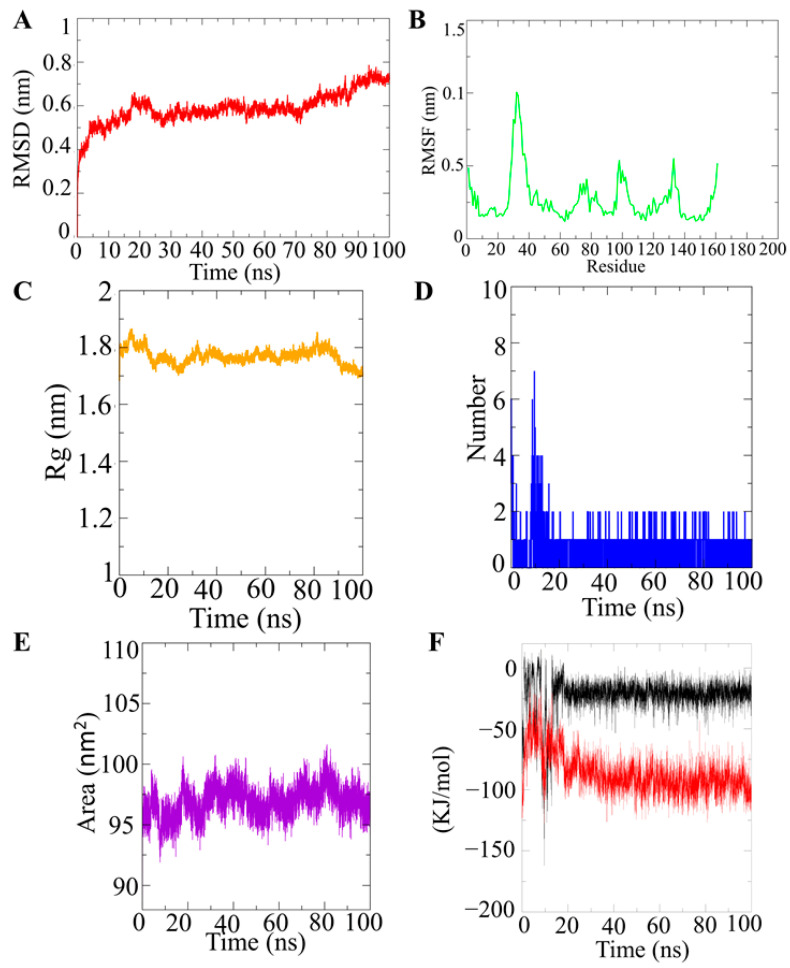
MD simulation analysis of the VanZ–G3K ligand. (**A**): RMSD plot of the VanZ system in complex with G3K. (**B**): RMSF plot of the VanZ system in complex with G3K. (**C**): Rg plot of the VanZ system in complex with G3K. (**D**): Hydrogen bond dynamics between VanZ and the G3K ligand. (**E**): SASA plot for the VanZ system in complex with G3K. (**F**): Coul-SR and LJ-SR interaction energy plot for the VanZ system in complex with G3K. MD simulation was repeated three times.

**Figure 5 ijms-23-00097-f005:**
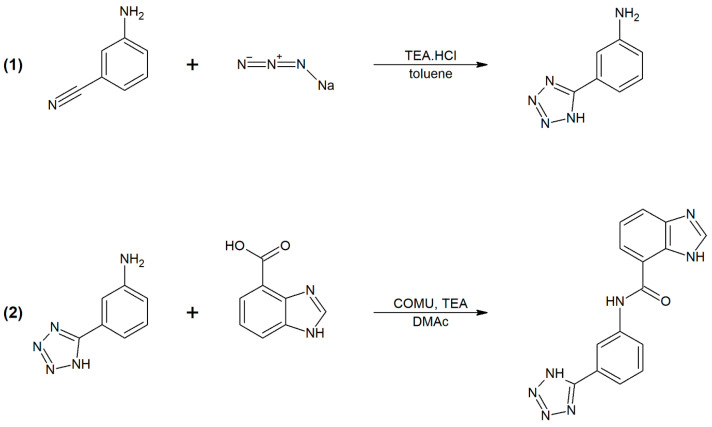
Two-step synthesis of G3K.

**Figure 6 ijms-23-00097-f006:**
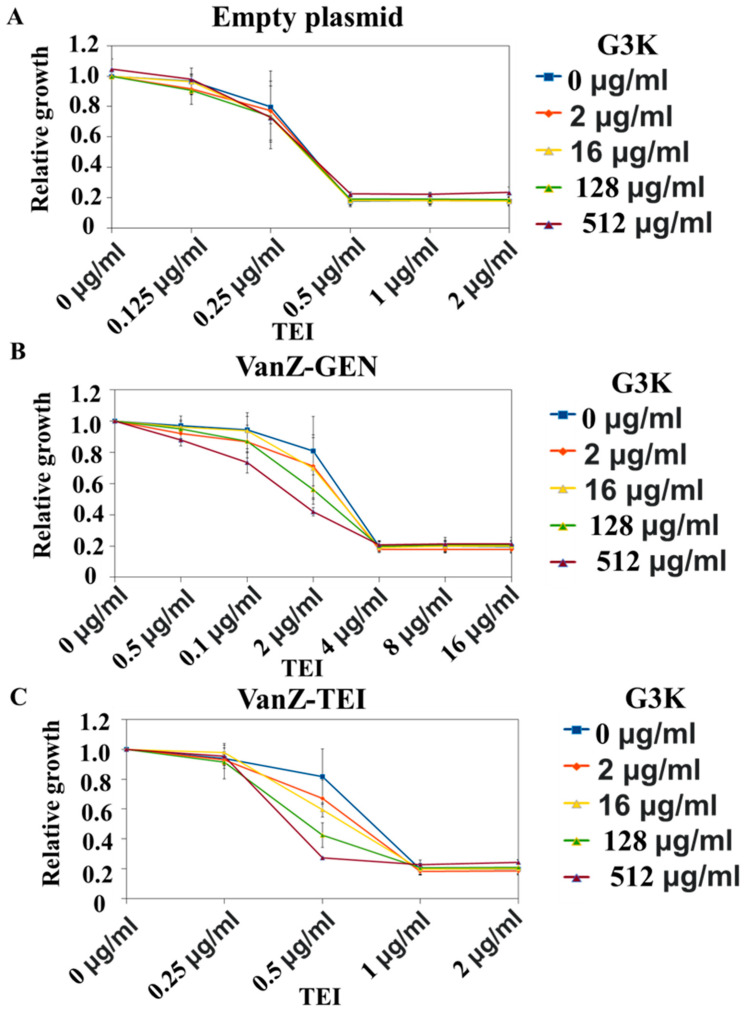
Effect of G3K on VanZ-mediated resistance to teicoplanin (TEI). (**A**): Where *S. aureus* RN4220 strain with empty plasmid was used. (**B**): Where *S. aureus* RN4220 strain heterogeneously expressing VanZ-GEN (*vanZ* encoded in the *E. faecium* genome) was used. (**C**): Where *S. aureus* RN4220 strain heterogeneously expressing VanZ-TEI (*vanZ* encoded in the *vanA* resistance gene cluster) was used. Data represent the mean ± SD, *n* = 3.

**Figure 7 ijms-23-00097-f007:**
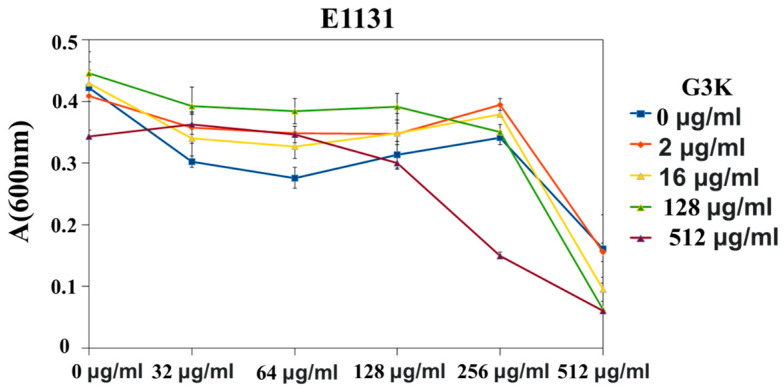
G3K effect evaluation on *vanA*-resistance-gene-cluster-mediated resistance to TEI in *E. faecium*. Data represent the mean ± SD, *n* = 3.

**Figure 8 ijms-23-00097-f008:**
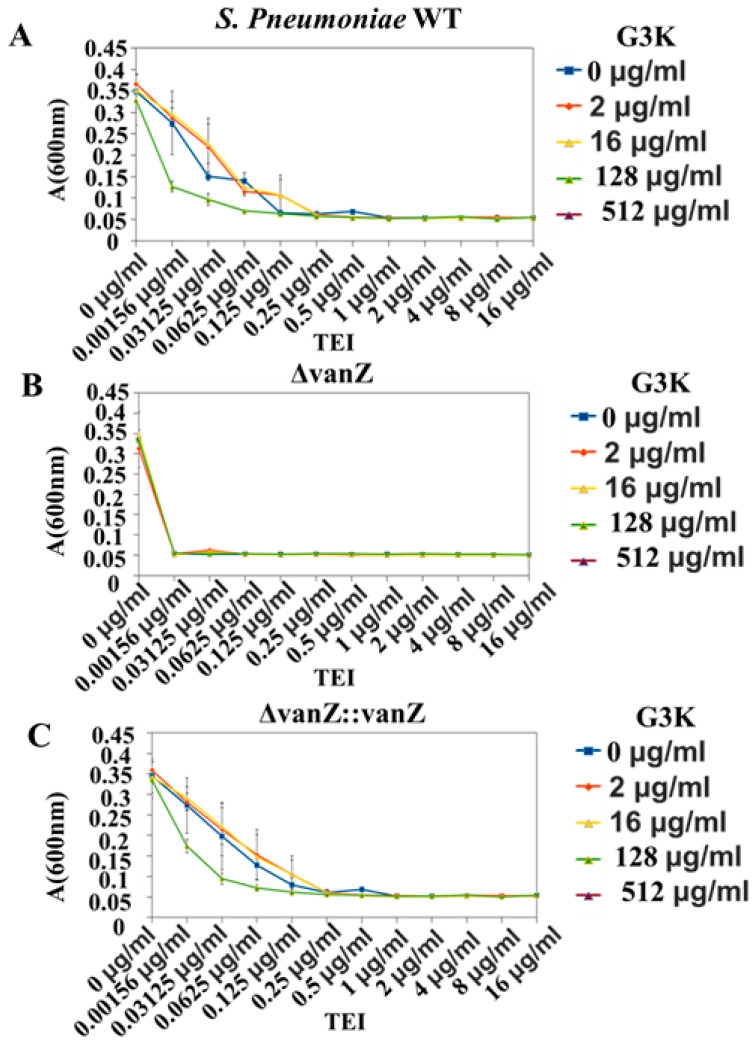
Effect of G3K on VanZ-mediated teicoplanin (TEI) resistance in *S. pneumonia R6*. (**A**): *S. pneumoniae* with knockout and complemented *vanZ* was used as the control. (**B**): Δ*vanZ* positive control strain was used, which showed high sensitivity towards TEI. (**C**): WTs and *vanZ*-reverted strains were used with G3K, where they increased the sensitivity of the strains to TEI. Data represent the mean ± SD, *n* = 3.

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
