# Peer review of "Specific Inhibition of VanZ-Mediated Resistance to Lipoglycopeptide Antibiotics"

_ijms, 2021, doi:10.3390/ijms23010097_

Round 1

Reviewer 1 Report

This is a well-written and well-presented manuscript that is scientifically sound. But I encourage to improve the language throughout the manuscript. For example, at the line 202, 'see Figure S9' should be in the parenthesis just like (Figure S9); it is not necessary to use the word 'see' to address the corresponding figure. Similar things appeared all over the manuscript which should be carefully addressed. Also, be careful to check if all the scientific names of bacteria are in their proper format (italic); for example 'S. pneumniae' should be italic which appeared at the top figure 8. 

Author Response

We thank the reviewer for these comments and suggestions. All the corrections were made very carefully according to the reviewer’s suggestions, figure 8 has been updated with proper “S. pneumoniae” in italics on the top of the figure. 

Reviewer 2 Report

Expression of vancomycin Z (vanZ) in several bacteria such as Staphylococcus aureus, Enterococcus faecium and Clostridium difficile, makes them less susceptible to several antibiotics including teicoplanin (TEI). In an effort to increase the antibiotic sensitivity of these bacteria, Authors, used computational methods and predicted a potential ligand G3K. G3K was synthesized, purified using HPLC and characterized using NMR. Authors demonstrate the G3K does not inhibit the bacterial growth but when supplemented with TEI increases the antibiotic sensitivity of bacteria. In brief, this is a complete study covering ligand prediction, synthesis, purification, and in-vitro functional characterization. I suggest following minor improvements.

  1. Please mention number of replicates in all experimental figures and type of error bar (STD or SEM).
  2. Correct the caption of figure 1.
  3. Caption of Figure 7 can be improved.
  4. It is advisable to include more intermediate data points in figure 6. Preferably where G3K starts showing its effect.

Author Response

  1. Please mention number of replicates in all experimental figures and type of error bar (STD or SEM).

We thank the reviewer for taking time to raise this important question. We performed all the in vivo experiments three times and data is displayed with SD error bars. These parts are updated in the manuscript with “Data represent the mean ±SD, n = 3.” All the in-silico work such as; docking, prediction, interaction, and MD simulation were also repeated three times and this is now also mentioned in the relevant sections.

  1. Correct the caption of figure 1.

We thank the reviewer for highlighting this issue, the figure 1 legend has been corrected to: “I-TASSER server based predicted structure of the VanZ protein”. The prediction was repeated in the same server three times. 

  1. Caption of Figure 7 can be improved.

We thank the reviewer for this valuable comment. The caption of figure 7 has been rephrased.

  1. It is advisable to include more intermediate data points in figure 6. Preferably where G3K starts showing its effect.

We thank the reviewer for raising this point. We tried several times previously to check the effects of the different concentrations however they did not show any significant changes and that is why we decided to exclude the data points from the figure. We felt that as the data points had no beneficial effect within the figure, we felt that including this data made the figure too cluttered and became more difficult for readers to understand.